# Towards Targeting Endothelial Rap1B to Overcome Vascular Immunosuppression in Cancer

**DOI:** 10.3390/ijms25189853

**Published:** 2024-09-12

**Authors:** Behshid Ghadrdoost Nakhchi, Ramoji Kosuru, Magdalena Chrzanowska

**Affiliations:** 1Versiti Blood Research Institute, Milwaukee, WI 53226, USA; bghadrdoostnakhchi@versiti.org (B.G.N.);; 2Department of Pharmacology and Toxicology, Medical College of Wisconsin, Milwaukee, WI 53226, USA; 3Cardiovascular Center, Medical College of Wisconsin, Milwaukee, WI 53226, USA; 4Cancer Center, Medical College of Wisconsin, Milwaukee, WI 53226, USA

**Keywords:** tumor angiogenesis, vascular immunosuppression, endothelial anergy, VEGF-A, small GTPase Rap1, cancer therapy

## Abstract

The vascular endothelium, a specialized monolayer of endothelial cells (ECs), is crucial for maintaining vascular homeostasis by controlling the passage of substances and cells. In the tumor microenvironment, Vascular Endothelial Growth Factor A (VEGF-A) drives tumor angiogenesis, leading to endothelial anergy and vascular immunosuppression—a state where ECs resist cytotoxic CD8^+^ T cell infiltration, hindering immune surveillance. Immunotherapies have shown clinical promise. However, their effectiveness is significantly reduced by tumor EC anergy. Anti-angiogenic treatments aim to normalize tumor vessels and improve immune cell infiltration. Despite their potential, these therapies often cause significant systemic toxicities, necessitating new treatments. The small GTPase Rap1B emerges as a critical regulator of Vascular Endothelial Growth Factor Receptor 2 (VEGFR2) signaling in ECs. Our studies using EC-specific Rap1B knockout mice show that the absence of Rap1B impairs tumor growth, alters vessel morphology, and increases CD8^+^ T cell infiltration and activation. This indicates that Rap1B mediates VEGF-A’s immunosuppressive effects, making it a promising target for overcoming vascular immunosuppression in cancer. Rap1B shares structural and functional similarities with RAS oncogenes. We propose that targeting Rap1B could enhance therapies’ efficacy while minimizing adverse effects by reversing endothelial anergy. We briefly discuss strategies successfully developed for targeting RAS as a model for developing anti-Rap1 therapies.

## 1. Introduction

The vascular endothelium consists of a specialized monolayer of endothelial cells (ECs) lining the interior surface of blood and lymphatic vessels, creating a barrier to control the movement of substances, cells, and pathogens from the bloodstream to the surrounding tissues. Under normal conditions, EC monolayers maintain a non-adhesive and quiescent state in non-inflamed tissues, playing a crucial role in preventing blood clotting, regulating blood flow, modulating fibrinolysis, and managing vessel wall permeability [1]. 

Under the conditions of hypoxia in the growing tumor, tumor blood vessels form from existing blood vessels through the process of angiogenesis. Induced by the hypoxic conditions of tumors, Vascular Endothelial Growth Factor A (VEGF-A) is the primary angiogenic factor responsible for the development of tumor vessels and is essential for tumor growth [2]. In response to VEGF-A, ECs drive this process, consisting of proliferation and migration, formation of vascular lumens, and construction of vascular networks [3]. Beyond supplying nutrients, tumor ECs influence the tumor microenvironment (TME). 

Inflammatory conditions induce expression of endothelial cell adhesion molecules (CAMs), such as vascular cell-adhesion molecule 1 (VCAM1) and intercellular adhesion molecule 1 (ICAM1), promoting leukocyte adhesion and transmigration [4]. In tumors, such activated ECs are crucial for initiation of leukocyte extravasation and infiltration, as well as orchestration of the inflammatory response. High leukocyte density in tumors, especially CD8^+^ cytotoxic T cells, is associated with reduced tumor growth and enhanced survival rates [5,6]. Thus, ECs play a pivotal role in modulating interactions with leukocytes and other components of the TME. However, the high content of angiogenic and proinflammatory cytokines in tumors leads to phenotypic, molecular, and functional alteration of the blood vessels. Tumor blood vessels display irregularity, fragility, and permeability, and the endothelial network is disorganized, with abnormal blood flow patterns and oxygen levels [3,7,8]. This aberrant morphology of blood vessels consequently results in hypoxic regions and acidic pH within the TME [9,10]. 

Increased expression of proangiogenic factors such as VEGF-A, fibroblast growth factor (FGF), and angiopoietin 1 leads to the phenotypic transformation of tumor ECs, a phenomenon called endothelial anergy, making them resistant to immune cell infiltration, particularly cytotoxic CD8^+^ T cells. This process involves the expression of immune inhibitory molecules (such as programmed cell death 1 (PD-1), PD-1 ligand1 (PD-L1), galectin-1, and FAS ligand (FASL)), loss of ability to upregulate CAMs, reduced adhesiveness, and insensitivity to inflammatory cytokines, thereby establishing a barrier to immune cell infiltration, in particular to activated cytotoxic CD8^+^ T cells—a condition known as tumor EC anergy [11,12,13,14,15,16]. Additional EC processes and molecules may contribute to the complexity of EC anergy. Accumulating evidence supports tumor EC senescence and the senescence-associated secretory phenotype in the modulation of tumor progression and immunotherapy [17,18]. In melanoma, autophagy has been proposed as a key tumor vascular anti-inflammatory mechanism dampening antitumor immunity [19]. In glioblastoma (GBM), a subset of tumor ECs, via a Twist1/SATB1-mediated sequential transcriptional activation mechanism, produces osteopontin to promote immunosuppressive macrophage (Mφ) phenotypes that inhibit T cell infiltration and activation [20]. Furthermore, VEGF-A prevents dendritic cell maturation and recruits a multitude of immune-suppressive cells, including myeloid-derived suppressor cells, Tregs, and M2 macrophages [21,22]. Mechanistically, VEGF-A inhibits EC activation by blocking tumor necrosis factor-α (TNF-α)-induced nuclear factor κ-light chain enhancer of activated B cells (NF-kB) activation, subsequently downregulating the expression of chemokines such as CXCL10, CXCL11, and colony stimulating factor 2 and CAMs [23]. This leads to the suppression of leukocyte recruitment and activation, a process called vascular immunosuppression [14,24,25]. 

## 2. Anti-Angiogenesis to Normalize Vessels, Current Anti-Angiogenic Therapies

Anti-angiogenesis has been recognized as an effective antitumor strategy since its proposal by Judah Folkman [26]. This approach aims to normalize blood vessels by pruning abnormal vessels and reducing permeability, thus improving the delivery of oxygen and therapeutic agents to the tumor. Anti-VEGF therapies for cancer have been developed and are in clinical use. Currently approved anti-angiogenic drugs for cancer treatment are bevacizumab, ramucirumab, and aflibercept [2]. 

Bevacizumab is a monoclonal antibody that specifically inhibits VEGF by blocking its interaction with its receptor VEGFR2 [27]. This drug is used as a first-line treatment for non-small cell lung cancer (NSCLC), colorectal cancer (CRC), renal cell carcinoma (RCC), and metastatic ovarian cancer [27,28,29]. Ramucirumab is a neutralizing antibody that directly targets VEGFR2, which is primarily expressed on vascular ECs, blocking the binding of VEGF, VEGFC, and VEGFD to VEGFR2 [30]. Ramucirumab is recommended for the first-line treatment of NSCLC, gastric cancer, hepatocellular carcinoma (HCC), and metastatic CRC [30,31,32]. Aflibercept is a genetically engineered soluble receptor that consists of the immunoglobulin domains of both VEGFR1 and VEGFR2 fused to the Fc portion of IgG. It binds to and neutralizes VEGF ligands, including VEGF, placental growth factor (PlGF), and VEGF-B. Aflibercept was approved for second-line treatment of CRC [33]. Tyrosine kinase inhibitors (TKIs), such as apatinib, axitinib, sorafenib, and sunitinib, inhibit VEGFR signaling and are used as the first-line treatment for HCC, metastatic RCC, thyroid cancer, and pancreatic neuroendocrine tumor [2]. Due to their broad spectrum of targets, TKIs frequently display significant toxicity profiles, which limit their long-term clinical utility and typically preclude their combination with cytotoxic agents [2].

Anti-angiogenic therapies not only normalize blood vessels, but also affect immunotherapy outcomes by modulating the TME. By normalizing the vasculature, anti-angiogenic therapies improve immune cell infiltration and function within the tumor, potentially enhancing the efficacy of immunotherapies.

## 3. Combining Immunotherapy with Anti-Angiogenesis (Combination Therapy)

Immunotherapies are promising treatments in cancer, aiming to activate the immune system to combat cancer cells, often by blocking immune checkpoint molecules such as PD1, PD-L1, and cytotoxic T lymphocyte antigen 4 (CTLA4) [34]. Another strategy is cellular immunotherapy or adoptive cell therapy. This approach entails extracting immune cells from cancer patients and genetically modifying the T cells to express chimeric antigen receptors (CAR T cells), which are then reintroduced into the cancer patients [35]. However, the clinical effectiveness of immunotherapy is influenced by tumor ECs. The infiltration of leukocytes, particularly T cells, into the TME depends on EC permeability and adhesion and extravasation of immune cells across the tumor blood vessels. Poorly perfused, disorganized, and leaky tumor blood vessels hinder the infiltration of leukocytes into tumors [36,37]. 

Anti-angiogenic drugs have been shown to normalize blood vessels by counteracting VEGF-induced EC anergy, enhancing immune responses by promoting vessel maturation, and alleviating the immunosuppression caused by hypoxia and VEGF [38,39,40,41,42,43,44]. These therapies upregulate CAMs on tumor vasculature, leading to greater leukocyte infiltration [45,46]. Additionally, blocking VEGF-A and angiopoietin 2 increases endothelial PD-L1 expression [47,48,49]. Thus, disrupting VEGF-A signaling can normalize the tumor vasculature and renew the responsiveness of ECs to adoptive T cell therapy. Vascular normalization has been shown to polarize tumor-associated macrophages to a proinflammatory M1 phenotype and enhance T cell infiltration within the tumor [50,51]. Therefore, normalizing tumor vasculature by suppressing VEGF signaling may reverse vascular immunosuppression within the TME. Indeed, bevacizumab and sunitinib have been shown to increase dendritic cell activation [52], enhance cytotoxic T cell numbers and function [53], and reduce regulatory T cell [54] and myeloid-derived suppressor cell [54] populations.

## 4. Combination Therapy Enhances Clinical Effectiveness

Combining anti-angiogenic drugs with immunotherapy has shown promising results in clinical practice [55]. This combination aims to target both EC anergy and immune checkpoints to enhance antitumor immune responses. Blocking interactions involving immune checkpoint molecules improves immune response intensity and duration [56], as seen in the efficacy of anti-PD-1 antibodies in treating melanoma, RCC, and NSCLC by revitalizing cytotoxic CD8^+^ T cells [56,57]. Immuno-stimulation using anti-angiogenic treatment has the potential to sensitize tumors with low PD-L1 expression to therapies targeting immune checkpoint inhibitors (ICIs) [48]. The heightened infiltration of T cells following anti-angiogenic treatment has been linked to an increase in PD-L1 expression within the TME [47,48]. Anti-angiogenic immune-modulating therapies induce the transdifferentiation of postcapillary venules into inflamed high-endothelial venules (HEVs) via lymphotoxin/lymphotoxin beta receptor (LT/LTβR) signaling. These tumor HEVs enhance intratumoral lymphocyte influx and create niches that support the differentiation of CD8 T cell progenitors into effector cells. The maintenance of tumor HEVs relies on continuous signals from CD8 and NK cells, indicating a feed-forward loop driven by the adaptive immune system [58]. 

Combining a mouse anti-VEGFR2 antibody or TKIs with antibodies that block PD1 or PD-L1 has been effective [36,59]. Major groups of approved ICIs include anti-PD-1 antibodies (pembrolizumab, nivolumab, camrelizumab), anti-PD-L1 antibodies (atezolizumab, avelumab), and anti-CTLA4 antibodies (ipilimumab). The FDA has recently approved combined therapy of anti-angiogenic drugs with ICIs for treating various cancers, including HCC, RCC, endometrial carcinoma, and NSCLC [36,60]. These beneficial interactions were also observed in breast cancer, pancreatic cancer, RCC, and glioblastoma multiforme [36,59]. Additionally, the combination of bevacizumab and atezolizumab has shown promising clinical benefits, achieving overall response rates of 11% to 50% with a manageable safety profile in advanced HCC patients [61]. Phase III studies have established the combination of axitinib with pembrolizumab (Keytruda) or avelumab (Bavencio) as the standard first-line treatment for RCC, significantly improving overall response rates by 55% in advanced RCC patients [62]. Therefore, tumor ECs play a critical role in regulating immune cell infiltration within tumors, improving the positive outcomes of immunotherapy. Overcoming EC anergy and promoting vascular normalization are promising strategies to enhance immune-mediated tumor destruction and improve therapeutic outcomes in cancer treatment.

## 5. Challenges of Combination Therapy, and the Need for New Targets

Combining anti-angiogenic drugs with immunotherapies holds promise for cancer treatment, but systemic administration of these drugs affects all tissues and organs due to their widespread distribution. This can lead to adverse effects, as anti-VEGF drugs induce vascular regression and alter vascular structures in endocrine organs, liver, bone marrow, and the gastrointestinal wall, which rely on VEGF for vascular integrity and homeostasis [2]. Continuous treatment with anti-angiogenic drugs, such as bevacizumab, however, can induce rebound proliferation of tumor blood vessels, leading to drug resistance. This occurs via TNF-α-induced expression of endothelial cell-specific molecule-1 (ESM1), which regulates matrix metalloproteinase 9, VEGF, and delta like canonical Notch ligand 4 (DLL4), thereby contributing to tumor metastasis [63]. Similarly, administering TKIs systemically in non-tumor mice results in significant microvessel regression in the thyroid, reducing thyroid hormone production [64,65]. Hypothyroidism is a common adverse effect in TKI-treated cancer patients [66]. Other adverse effects include hypertension, proteinuria, hemorrhage, and gastrointestinal perforation, although these are less severe with more selective agents like antibodies [67]. These side effects limit the clinical therapeutic efficacy of anti-angiogenic drugs, highlighting the need for new targets to improve effectiveness while minimizing adverse effects [55].

The variability in the vascular normalization window across cancers complicates treatment. Combining anti-angiogenic agents with ICIs like anti-PD1 or PD-L1 can enhance delivery by normalizing vessels [55,68]. Mathematical models support simultaneous antibody administration for better vessel perfusion [69]. Clinical trials with bispecific antibodies, such as ivonescimab in advanced NSCLC, have shown promising activity and tolerable safety [68]. However, combining anti-angiogenics with other targeted antibodies, like cetuximab or trastuzumab, may reduce their levels in tumors [70,71,72]. Furthermore, the efficacy of combination therapies is further influenced by sensitivity to monotherapy (either anti-angiogenesis or ICIs), which varies across different types. According to hypothetical Kaplan–Meier survival curves from phase III clinical trials, cancers sensitive to anti-angiogenic agents, such as HCC and RCC, are more likely to derive substantial benefit from combination therapies with ICIs [55,68,73,74]. In contrast, cancers like NSCLC and ovarian cancer, which are less sensitive to anti-angiogenic agents as monotherapy, may show mixed or minimal additional benefit from combination with ICIs [61,75]. In some cases, such as metastatic CRC, combining anti-angiogenic therapies with ICIs has not shown significant survival benefits compared to anti-angiogenic therapy alone [76]. This variability in response suggests that VEGF–VEGFR-mediated immunosuppression affects ICI resistance differently across cancer types, underscoring the need for tailored treatment strategies based on the specific drug sensitivity profile of each cancer type [55].

One emerging target for anti-angiogenic therapy in cancer is the small GTPase Rap1B (Figure 1). Earlier studies implicated both Rap1 isoforms in adhesion and migration via integrins [77,78,79]. Accumulating evidence shows that, in ECs, Rap1B is a positive regulator of VEGFR2. Recent studies show that Rap1B in tumor endothelium mediates VEGF-mediated vascular immunosuppression, raising the exciting possibility of being a novel target in anti-angiogenesis therapy, as detailed in the next section.

## 6. Rap1: A Ras Family Member

Rap1, a member of the Ras superfamily and a close relative of RAS, shares structural similarities with RAS proteins. The Ras isoforms—H-RAS, N-RAS, K-RAS4A, and K-RAS4B—act as signaling nodes activated by extracellular stimuli to regulate gene expression, cell proliferation, differentiation, and survival through interactions with downstream effectors. Mutated in many cancers, RAS is an oncoprotein crucial in human cancer development [80,81]. In contrast, the best characterized function of Rap1, particularly in vitro, is the positive modulation of integrins and cadherins [82]. In vivo and mechanistic studies have revealed important functions of Rap1 in ECs, including the control of VEGFR2 signaling, as described below.

The RAS proteins function as GDP/GTP-regulated molecular switches, where GTP binding and hydrolysis are fundamental to their biological function, with the GTP-bound form facilitating interactions with downstream effector proteins [83]. The activity of RAS and Rap1 is controlled by guanine nucleotide exchange factors (GEFs), which facilitate the exchange of GDP for GTP, thereby activating them, and by GTPase-activating proteins (GAPs), which enhance their inherent GTPase activity, causing hydrolysis of GTP to GDP, thus inactivating them [83,84,85].

Key molecular features of Rap1 and Ras include the conserved “G box” sequences (G1, G2, G3, G4, and G5), which are involved in GTP binding and hydrolysis, and effector-binding regions that interact with RA domains or RBDs in various proteins. The effector regions consists of switch I and switch II loops, whose conformation is modulated by GTP binding (Figure 2) [83,86].

Signaling by Rap1 and RAS is influenced by the C-terminal hypervariable region (HVR) and post-translational lipid modifications of the terminal CAAX motif (C: cysteine, A: aliphatic amino acid; X: any amino acid), which is crucial for their subcellular localization and function [88,89,90,91]. CAAX box processing involves a three-step maturation mechanism: farnesylation (RAS) or geranylgeranylation (Rap1) of the cysteine, proteolytic cleavage of the last three amino acids (AAX), and carboxyl methylation, which are required for stable membrane localization [88,92]. Reversible palmitoylation of the C-terminal cysteine of RAS proteins further controls its trafficking between the plasma membrane and the Golgi apparatus [88,93]. The sequences within HVR are involved in specific lipid binding [94,95]. These modifications facilitate RAS and Rap1 localization to the inner leaflet of the plasma membrane, resulting in diverse effector contacts and signaling outputs [91,96].

## 7. Ras and Rap1 Effector Domains and Direct Targets

Rap1 shares structural similarities with RAS proteins, particularly in the switch I region, which is identical in both proteins, indicating a high degree of conservation. However, the switch II regions shows significant differences, leading to variations in their specific interactions with other proteins or regulatory mechanisms [97,98]. These differences in the switch II region could account for the distinct regulatory functions and interactions of Rap1 compared to RAS, despite their shared mechanism of action involving nucleotide-mediated conformational changes [98].

The best-known pathway involves RAS activation by the epidermal growth factor receptor, leading to a signaling cascade through Raf kinase, MAP kinase/ERK1 kinase (MEK1/2), and extracellular signal-regulated kinase 1/2 (ERK1/2) that plays a key role in controlling cell proliferation and survival [83,99]. While Rap1 can also activate ERK1/2, it often does so synergistically with RAS. For example, in response to nerve growth factor stimulation, Rap1 activation stimulates ERK via B-Raf, which is initially triggered by RAS [100,101]. In ECs, Rap1 plays a critical role in promoting proliferation and migration, partly via the ERK pathways [79,102]. However, direct Rap1 effectors modulate integrins and cadherin, thus regulating cell adhesion, junction formation, and polarity [82,100]. In ECs, Rap1 modulates adhesion to the extracellular matrix, migration, tube formation, and signaling via integrins [77,79,102,103,104]. In addition to distinct effectors, RAS and Rap1 function in distinct signaling networks due to their interactions with different sets of GEFs and GAPs specific to each network [100].

## 8. Rap1 Is a Positive Regulator of VEGF Signaling

Rap1 is essential for angiogenic responses in vitro, to VEGF-A and basic FGF, and for angiogenic neovascularization in vivo. Rap1B-deficient mice exhibit inhibited proliferation of retinal ECs, impaired migration of lung ECs, and decreased microvessel sprouting in response to VEGF-A and basic FGF in vitro. Rap1B deficiency leads to decreased activation of p38 MAPK and p42/44 ERK following VEGF-A stimulation [78]. Additionally, key signaling molecules of angiogenesis, such as ERK, p38, and Rac, are reduced in response to FGF2 in the absence of either of Rap1 isoforms [79]. Deletion of both Rap1A and Rap1B also leads to decreased phosphorylation of tyrosine 397 in focal adhesion kinase (FAK) and blocks VEGF-induced activation of Akt1 [77].

In vivo, Rap1B deficiency results in impaired neonatal retinal angiogenesis, which is dependent on VEGF-A, as well as reduced VEGF-induced Matrigel plug neovascularization, a model of tumor angiogenesis [78]. Additionally, Rap1 is required for FGF-induced Matrigel neovascularization [77,79]. Using EC-specific knockout mice, we demonstrated that Rap1B-dependent angiogenesis is EC-endogenous [103]. In a zebrafish model of VEGF-dependent angiogenesis, we demonstrated that Rap1B is essential for the formation of intersomitic vessels, acting in the same pathway as VEGFR2 [103]. Mechanistically, Rap1B knockout leads to impaired VEGF-induced VEGFR2 phosphorylation, indicating that Rap1B promotes VEGF signaling by enhancing VEGFR2 kinase activity, partly mediated by integrin α_v_β_3_ [103].

These findings highlight Rap1 as positive regulator of VEGFR2 activity (Figure 1A), suggesting broader implications for Rap1 as a mediator of VEGF-dependent effects on ECs. Indeed, Rap1B is required for VEGF-dependent EC junction dissolution in vitro, as well as in vivo under conditions of excessive VEGF in early diabetes, where EC-Rap1B deletion protects against retinal vessel hyperpermeability [104]. Rap1B is also necessary for VEGFR2 transactivation, which is required for endothelial nitric oxide (NO) synthase (eNOS) activation and NO release in response to shear flow [105]. These findings underscore the central role of Rap1 as a positive regulator of VEGFR2 activation and signaling, prompting further examination of its role in VEGF-dependent tumor angiogenesis.

## 9. Role of Rap1B in VEGF-A-Induced Immunosuppression in the TME

Endothelial Rap1B is emerging as a key player in cancer progression, as suggested by several findings. When we examine the expression of Rap1B in ECs from tumors in lung, colorectal, and ovarian cancer patients using publicly available single-cell RNA sequencing (scRNAseq) data, we find that its levels are notably higher in tumor ECs compared to their non-tumor counterparts [106,107]. This raises the possibility that EC Rap1B contributes to tumor growth. To delve deeper, we employed a melanoma model that is heavily reliant on angiogenesis for tumor growth. In this model, we used EC-specific Rap1B knockout (Rap1B^iΔEC^) mice and found that the absence of EC Rap1B led to a significant reduction in tumor growth. Additionally, the tumor ECs in these knockout mice exhibit altered vessel morphology, with fewer large blood vessels (Figure 1B) [107]. Further investigation revealed that the deletion of EC-specific Rap1B significantly impacted the TME. There is a notable increase in infiltration of CD45^+^ leukocytes and enhanced activation of CD4^+^ and CD8^+^ T cells from Rap1B^iΔEC^ tumors. Interestingly, when CD8^+^ T cells are depleted, tumor growth is normalized in Rap1B^iΔEC^ mice, indicating that Rap1B in tumor ECs plays a crucial role in controlling CD8^+^ T cell activity [107].

Moreover, Rap1B-deficient ECs show an increased response to TNF-α, a cytokine associated with the TME, which leads to heightened T cell adhesion. Transcriptomic analysis further supports these findings, revealing a notable increase in TNF-α signaling pathways and NF-κB transcription in the absence of Rap1B. This is accompanied by a rise in proinflammatory CAMs in Rap1B-deficient ECs [107,108]. Additionally, Rap1B deficiency results in elevated expression of chemokines, including CXCL11, and CAMs following TNF-α stimulation [107]. These results suggest that Rap1B plays a role in limiting leukocyte recruitment and modulating the interactions between ECs and leukocytes, both in vitro and within tumor ECs in vivo.

In the TME, VEGF, which is typically elevated, induces immunosuppression in ECs by interfering with TNF-α-induced and NF-kB-mediated expression of CAMs, thereby inhibiting leukocyte adhesion, particularly of CD8^+^ T cells [23,44]. Interestingly, this immunosuppressive effect of VEGF-A is significantly diminished in Rap1B-deficient ECs. Even when Rap1B-deficient tumor ECs are treated with TNF-α, VEGF-A cannot suppress TNF-α-induced expression of ICAM1 or VCAM1.

These findings collectively underscore the critical role of Rap1B in the immunosuppressive signaling pathways mediated by VEGF-A that lead to EC anergy to proinflammatory cytokines. By promoting angiogenesis and inhibiting leukocyte recruitment, tumor EC Rap1B significantly influences EC responses in the TME (Figure 1C) [107]. Therefore, targeting Rap1B signaling in tumor ECs could offer a promising strategy to counteract EC anergy in cancer therapy (Figure 1A,B).

## 10. RAS: Now Druggable—Paving the Way for Targeting Rap1?

Our preclinical study identified EC Rap1B as a potential novel antitumor target [107]. Currently, there are no specific anti-Rap1 treatments. However, Rap1 is structurally close to the oncogene RAS. Historically, targeting RAS has posed substantial challenges, due to its structure lacking well-defined binding pockets for small molecule inhibitors [109,110]. Recently, RAS has been successfully pharmacologically targeted, and some of the approaches used to target RAS may inform developing anti-Rap1 treatments [87,111].

Mutations in the Ras gene, occurring in approximately 25% of human cancers, play crucial roles in driving tumor initiation and progression [81,112,113]. Ras mutations typically involve single base missense mutations, predominantly affecting residues G12, G13, or Q61 within the switch I and switch II regions. These mutations disrupt GAP-mediated GTP hydrolysis, leading to accumulation of RAS-GTP and elevated signaling to ERK and PI3K [114,115].

Ras mutations are not evenly distributed among different cancer types, with Kras mutations being the most prevalent (86%), followed by Nras (11%) and Hras (3%) mutations [114]. Specifically, Kras mutations are almost ubiquitous in pancreatic ductal adenocarcinoma (PDAC) and lung adenocarcinoma (LAC) [116]. In CRCs, Kras mutations are predominant, whereas Nras and Hras mutations are rare [117]. In multiple myeloma (MM), both Kras and Nras mutations occur at similar frequencies, whereas Nras mutations predominate in cutaneous melanomas and acute myelogenous leukemias (AMLs). Hras mutations are most commonly observed in bladder carcinomas and head and neck squamous cell carcinomas (HNSCCs) [114].

## 11. Strategies for Targeting RAS-Mutant Tumors and Use of Drugs in Cancer Treatment

### 11.1. Switch II Pocket Inhibitors, KRAS-G12C

KRAS is essential for development in mice [118,119]; thus, targeting WT KRAS raises toxicity concern. To circumvent potential toxicity in humans, the field of RAS covalent inhibitors, pioneered by Shokat, has focused on targeting the KRAS-G12C mutant. This approach leverages the cysteine residue at position 12, which is unique to the G12C mutant and absent in WT KRAS. Shokat’s team discovered a new allosteric binding pocket behind switch II, named the switch II pocket, in the KRAS-G12C mutant protein and developed compounds that irreversibly bind KRAS-G12C, resulting in its allosteric inhibition by locking KRAS-G12C in an inactive GDP-bound conformation (Figure 2A) [120].

More potent switch II pocket inhibitors have been developed by pharmaceutical companies. Notably, AMG-510, also known as sotorasib (Lumakras), is the first FDA-approved KRAS-G12C inhibitor for treating NSCLC patients harboring this mutation [121,122,123,124]. MRTX849 (adagrasib, Krazati) developed through structure-based design, is a selective inhibitor of mutant KRAS cell growth and has been approved for treatment of NSLC and, in combination with cetuximab (anti-EGFR antibody), for treatment of metastatic CRC [125]. By inhibiting EGFR in addition to KRAS-G12C, a more comprehensive suppression of oncogenic RAS signaling can potentially lead to enhanced therapeutic benefits [126]. The switch II pocket is highly dynamic, as evidenced by molecules that bind a new groove (switch II groove), adjacent to the switch II pocket but away from the nucleotide-binding site [127]. Importantly, these molecules bind both GDP-bound and GTP-bound KRAS (Figure 2B) [127]. Additional allele-specific inhibitors of KRAS are being developed and are showing clinical promise [109].

Strategies developed for targeting KRAS-G12C, particularly covalent inhibitors and specific allosteric inhibitors, are highly specific to KRAS. Rap1 is seldom mutated, and the G12C mutation has not been reported; thus, this approach is unlikely to be effective for Rap1.

### 11.2. Direct Targeting of WT, G12C KRAS

The above-described inhibitors target the inactive, GDP-bound state of mutant KRAS, which makes them vulnerable to mechanisms of resistance that increase levels of GTP-bound KRAS or wild-type HRAS and NRAS. Furthermore, the KRAS-G12C mutation is only present in a small fraction of cancers [87]. Recent breakthroughs include RMC-7977, a reversible, tri-complex RAS inhibitor with broad-spectrum activity for the GTP-bound WT and mutant KRAS, NRAS, and HRAS variants [128]. RMC-7977 acts via non-covalent inhibition by forming a tri-complex with cyclophilin A (CYPA), which sterically occludes RAS–effector interactions (Figure 2B) [129]. Preclinical studies demonstrated that RMC-7977 showed strong effectiveness against RAS-addicted tumors with various RAS genotypes, especially KRAS codon 12 mutations (KRAS-G12X) [128,130]. RMC-7977 also inhibited the growth of KRAS-G12C models resistant to KRAS-G12C inhibitors by restoring RAS pathway signaling [128]. RMC-7977 caused apoptosis and sustained growth arrest in tumors, but only temporarily reduced proliferation in normal tissues without apoptosis. In the KPC mouse PDAC model, RMC-7977 extended survival, with relapse linked to Myc copy number gain, which could be countered by combining treatment with TEAD inhibition in vitro [130].

While RMC-7977 holds high promise in targeting RAS, it is unlikely to inhibit Rap1. Rap1 structure is significantly different from RAS structure, and it does not bind CYP1, a prerequisite for this mechanism. Due to these differences, RMC-7977 is unlikely to be useful for targeting Rap1.

### 11.3. DCAI Pocket Binders: Inhibiting Ras–Effector Binding

Through an NMR-based fragment screen, Maurer et al. identified compounds, including DCAI, that bind to a common site of RAS, a pocket on the RAS protein that is adjacent to the switch I/II regions. Crystallography revealed that DCAI binds a pocket between the α2 helix and the core β-sheet, β1–β3 (DCAI pocket; Figure 2C), blocking the interaction between RAS and its GEF, SOS1, thus inhibiting KRAS activation [131]. Additional compounds binding to this pocket inhibit interaction with SOS1 or with RAS effectors [132,133]. Conceivably, this approach could also be used to target Rap1, but determining the exact effectors of Rap1 in the context of VEGFR2 signaling is required.

### 11.4. Indirect Targeting of Ras

Indirect targeting strategies focus on essential steps of RAS activation: nucleotide exchange, processing, membrane localization, and effector binding to inhibit RAS signaling indirectly.

### 11.5. SOS Inhibitors

SOS is a RAS GEF necessary for RAS-GTP loading and downstream signaling. Synthetic α-helical peptides that interfere with SOS-mediated RAS activation and function have been described [134,135]. Small molecule inhibitors targeting the CDC25 domain of SOS1 a region adjacent to switch II on RAS in the RAS–SOS1–RAS ternary complex have shown promise (Figure 2D) [136,137,138]. BI-1701963, a compound that inhibits SOS1, is currently in phase I clinical trials, being evaluated both as a standalone treatment and in conjunction with the MEK inhibitor trametinib [87].

Applicability of Rap1: The general concept may be adaptable to Rap1. Understanding the GEFs and GAPs specific to Rap1 in the context of VEGFR2 signaling could inform the development of small molecule inhibitors that disrupt Rap1 activation and function.

### 11.6. SHP2 Inhibitors

SHP2 is a non-receptor protein tyrosine phosphatase, presumed to act as a scaffold and required for full activation of the MAPK pathway downstream from RAS-GTP [132]. SHP2 allosteric inhibitors have been developed and are currently in clinical trials, showing potential in impacting downstream signaling and tumor growth [133,134]. While the role of SHP-2 in Rap1 activation has not been explored, the Rap1 effector afadin has been linked with regulation of SHP-2 activity and consequences for ERK signaling [135,139,140,141,142]. Further investigation into its biology is warranted.

### 11.7. RAS-Mimetics: Inhibition of Ras–Effector Interactions

RAS mimetics are small molecule inhibitors that bind to the RAS-binding domain (RBD) of RAS effectors, preventing their interaction with RAS. Examples include rigosertib, which binds to the RBDs of RAF kinases, Ral-GDS, and PI3K [143] and interferes with RAS-dependent tumor cell line growth [144]. Similarly, synthetic peptide inhibitors designed to imitate RAS RBDs have been shown to inhibit signaling downstream from RAS and decrease pancreatic cancer cell line survival [145].

While Ras mimetics would likely not inhibit all Rap1 signaling, the general approach could also be utilized to block Rap1 in tumor vessels.

## 12. Targeting Post-Translational Modification of RAS

RAS protein activity depends on post-translational lipid modifications of the CAAX motif, essential for membrane localization and for RAS protein function [91]. Blocking these post-translational modifications hinders RAS attachment to the cell membrane and subsequent activation of RAS signaling pathways [87]. Inhibitors targeting the three enzymatic post-translational processing steps have been developed: CAAX cysteine prenylation; cleavage of the terminal AAX residues by RAS-converting enzyme (RCE1); and methylation of the cysteine residue of the CAAX box by isoprenylcysteine carboxyl methyltransferase (ICMT).

### 12.1. Farnesyltransferase Inhibitors (FTIs)

HRAS undergoes exclusive prenylation by farnesyltransferase (FTase), unlike KRAS and NRAS, which can also undergo geranylgeranylation in the presence of FTIs, making FTIs less effective for these RAS isoforms [90,146]. FTIs could potentially be beneficial in the treatment of HRAS-mutant malignancies. Responses to the FTI tipifarnib (Figure 2F) were reported in patient-derived models of HRAS-mutant head and neck squamous cell carcinoma (HNSCC) and NSCLC [147].

### 12.2. Postprenylation CAAX Box Processing

Postprenylation CAAX box processing by Ras converting enzyme 1 (RCE1) and isoprenylcysteine carboxyl methyltransferase (ICMT) also contributes to RAS membrane localization and may offer additional targets. Several inhibitors have shown biological effects in preclinical models [148,149,150]. However, challenges remain, such as unclear requirements for RCE1 and ICMT in oncogenic RAS functions, confusing outcomes from genetic validation studies, lack of understanding of RCE1 CAAX protease activity mechanisms, toxicity, and difficulties in creating potent and selective inhibitors of these enzymes for clinical use [90,109,151].

### 12.3. Prenyl-Binding Protein Inhibitors

Interactions with prenyl-binding guanine nucleotide dissociation inhibitor-like proteins such as galectin-1 (HRAS, [152]), galectin-3 (KRAS, [153]) and phosphodiesterase-δ (PDEδ, [154]), play critical roles in trafficking Ras. They act as chaperones, transporting RAS proteins from endomembranes through the hydrophilic cytoplasm to the plasma membrane [90,155]. PDEδ plays an essential role in localizing both RAS and Rap1 [154,156]. Inhibitors like deltarasin, which disrupt KRAS-PDEδ binding, prevent KRAS signaling by mislocalizing KRAS [154,155,157,158]. Another compound, (E)-N’-((3-(tert-butyl)-2-hydroxy-6,7,8,9-tetrahydrodibenzo[b,dfuran-1-yl)methylene)-2,4-dihydroxybenzohydrazide (NHTD), similarly inhibits KRAS signaling and NSCLC growth in a preclinical model [87,159]. A spiro-cyclic PDEδ inhibitor, compound 36l, displays potent antitumor activity both in vitro and in vivo [160].

Inhibition of the Rap1–Pde6δ interaction has also been demonstrated as an effective strategy for inhibiting Rap1 signaling. The tryptamine-derived small molecule compound REM, which binds the prenyl-binding pocket of Pde6δ, disrupts Rap1 signaling [161]. This approach shows promise for inhibiting Rap1 in tumor vessels.

### 12.4. Geranylgeranyltransferase Inhibitors (GGTIs)

Unlike RAS, which is farnesylated, Rap1 undergoes geranylgeranylation [162]. Therefore, while FTIs would not be suitable for targeting Rap1, GGTIs might be considered [162]. GGTIs block the activity of GGTase-I, preventing the geranylgeranylation of target proteins, including Rap1 [163]. GGTIs have been used to inhibit Rap1 [164,165]. However, this approach lacks specificity, as GGTIs would target multiple geranylgeranylated proteins, including Rho family members, that are essential for cell signaling, proliferation, survival, cytoskeletal organization, cell migration, invasion, and membrane trafficking [166,167,168]. Nonetheless, such an approach might be advantageous in oncogenic pathways involving multiple GG-modified proteins [169].

### 12.5. Ras Oligomerization

Post-translationally modified RAS localizes to the inner leaflet of the PM, where it oligomerizes and forms nanoclusters, essential for RAS signaling [90,91]. O’Bryan’s team described a functionally critical region of RAS located outside of the effector lobe that is essential for nanoclustering: the α4–β6–α5 interface [170]. Using a monobody approach, which utilizes synthetic proteins constructed based on the molecular scaffold of the fibronectin type III domain, they developed a high-affinity binding protein, the NS1 monobody, that selectively bound GTP, GDP-H, and K-RAS (Figure 2G). Binding of the NS1 monobody disrupts HRAS dimerization, inhibiting HRAS-driven signaling pathways and providing a novel approach to targeting RAS activity [171,172]. O’Bryan’s group has utilized monobody technology to provide tools for understanding RAS function and biology in cells and to develop specific protein-based inhibitors for anti-RAS therapeutics [172,173,174]. This approach is based on the development of small single-domain antibodies that can be molecularly engineered to bind proteins inside or outside of cells with high affinity and selectivity.

Given the similarities between RAS and Rap1, strategies that target RAS nanoclustering and oligomerization may offer new insights into Rap1 biology, which is less well understood compared to RAS. By targeting regions of Rap1 critical for its function, similar to how the NS1 monobody disrupts HRAS dimerization, it may be possible to inhibit Rap1-dependent signaling pathways involved in VEGF-induced vascular immunosuppression and angiogenesis. The success of single-domain antibodies in clinical trials highlights their potential for targeting intracellular proteins like Rap1 [175].

## 13. Inhibition of RAS Signaling Pathways

Two main strategies to target the RAS pathways include identifying genes that are synthetically lethal with RAS mutations and targeting tyrosine kinase receptors (EGFR family) and RAS effector pathways: MAPK and PI3K [87].

### 13.1. Synthetic Lethal Screens

Synthetic lethal screens, complied in The Broad Institute dependency map (DepMap) portal, have identified RAF1 (CRAF) and SHOC2 as top essential genes in KRAS- and NRAS-mutant cell lines [176,177,178]. These genes represent potential therapeutic targets in RAS-mutant cancers.

### 13.2. EGFR Inhibition

EGFR inhibition as monotherapy has shown mixed results, but when combined with MEK or KRAS-G12C inhibitors it has shown promise [87]. This combination approach leverages the interaction between EGFR and downstream RAS signaling pathways to enhance therapeutic efficacy.

### 13.3. MAPK Pathway Inhibitors

Kinase inhibitors targeting BRAF-V600 and MEK are approved for BRAFV600-mutant metastatic melanoma but not for RAS-mutant tumors. These inhibitors effectively inhibit RAF monomers, as BRAF-V600 signals as a monomer. However, RAS-mutant tumors signal through BRAF and CRAF dimers, and BRAF-V600 inhibitors can paradoxically activate MAPK in RAS-mutant tumors [179,180]. RAF dimer inhibitors like belvarafenib show efficacy with minimal paradoxical activation [181].

MEK inhibitors demonstrate limited efficacy as monotherapies due to feedback mechanisms but exhibit synergistic effects when combined with RAF inhibitors. Combining PI3K and MEK inhibitors has shown preclinical success [182,183], but clinical trials face toxicity issues [184,185]. To address toxicity, researchers identified insulin-like growth factor 1 receptor (IGF1R) as the receptor tyrosine kinases that can suppress the PI3K and MAPK pathways [186,187]. Effective treatment of RAS-mutant tumors likely requires combining inhibitors targeting multiple nodes of the MAPK and PI3K pathways to achieve sustained suppression [87].

MAPK and PI3K pathways are also relevant for Rap1 signaling in ECs [102]. Therefore, MAPK and PI3K inhibitors developed for RAS might also work for Rap1. However, Rap1 acts upstream from VEGFR2, through an only partly understood mechanism [103]. Further research is needed to understand this mechanism fully.

## 14. Conclusions and Future Directions

Combination therapies, especially those pairing anti-angiogenic drugs with immunotherapies, show promise in cancer treatment but highlight the need for new, more targeted agents. Rap1B, a close relative of RAS, is an essential regulator of VEGFR2 signaling in ECs. Elevated in tumor ECs, Rap1B promotes tumor growth by enhancing VEGFR2 activity and contributing to vascular immunosuppression, which impedes immune cell infiltration.

Targeting Rap1B offers a novel strategy to overcome EC anergy and improve cancer treatment outcomes. Strategies successfully developed for RAS, such as interfering with GEFs, effectors, and protein oligomerization, could inform the development of Rap1B inhibitors. These inhibitors aim to disrupt Rap1B function, normalize tumor vasculature, and enhance immune cell infiltration, ultimately leading to more effective cancer therapies.

Further research is necessary to understand the molecular details and mechanisms of Rap1B activation, including identifying specific GEFs and elucidating Rap1B biology. Approaches utilizing single-chain antibodies, which have proven effective in studying RAS, could provide new tools to study Rap1B biology and aid in the development of targeted therapies. This knowledge will be crucial in developing specific Rap1B inhibitors and validating their clinical potential.

## Figures and Tables

**Figure 1 ijms-25-09853-f001:**
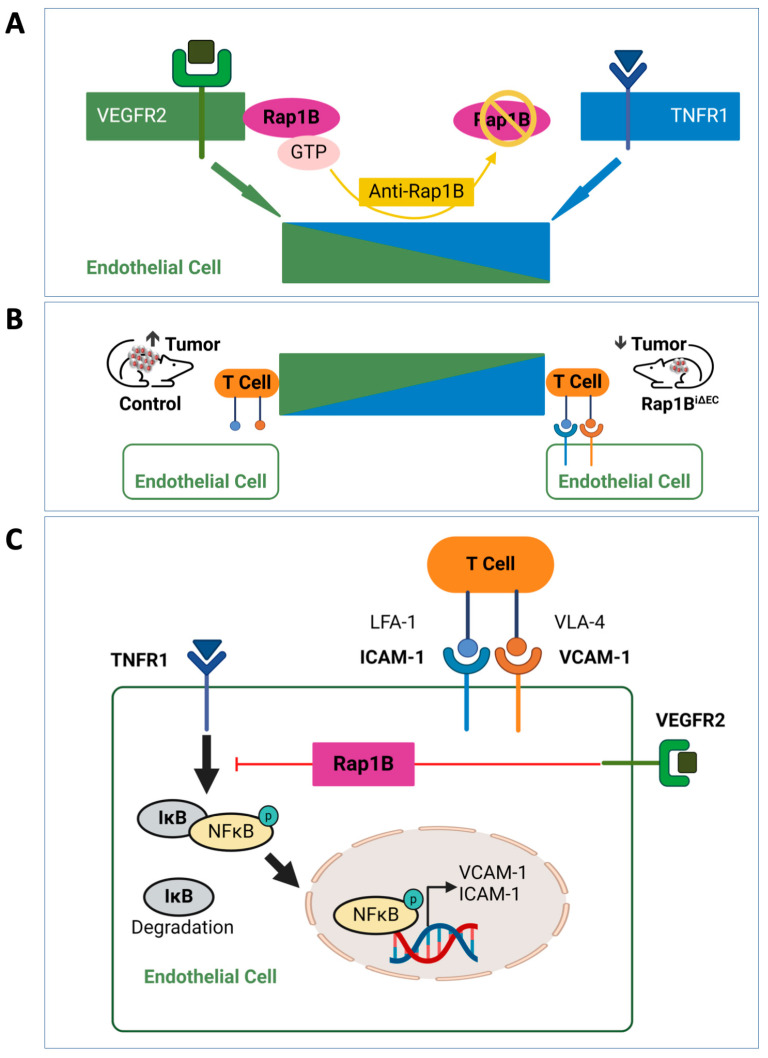
Targeting EC Rap1B to overcome VEGF-A-induced vascular immuno-suppression—a model. (**A**) Rap1B-GTP promotes Vascular Endothelial Growth Factor (VEGF) Receptor 2 (VEGFR2) signaling, inhibits proinflammatory signaling, and is a potential anti-cancer target. (**B**) Deletion of Rap1B in ECs inhibits tumor growth and promotes leukocyte infiltration (Rap1B^iΔEC^ mice). (**C**) Rap1B mediates VEGF-A-induced suppression of proinflammatory nuclear factor κ-light chain enhancer of activated B cells (NF-κB) signaling, including cell adhesion molecule (CAM) expression, limiting T cell adhesion and recruitment. ICAM—intracellular adhesion molecule; IκB—inhibitor of κB; LFA-1—lymphocyte function associated antigen 1 (integrin αLβ2); TNFR1—tumor necrosis factor receptor 1; VCAM—vascular CAM; VLA-4—very late antigen 4 (integrin α4β1). Signal transduction is indicated by arrows.

**Figure 2 ijms-25-09853-f002:**
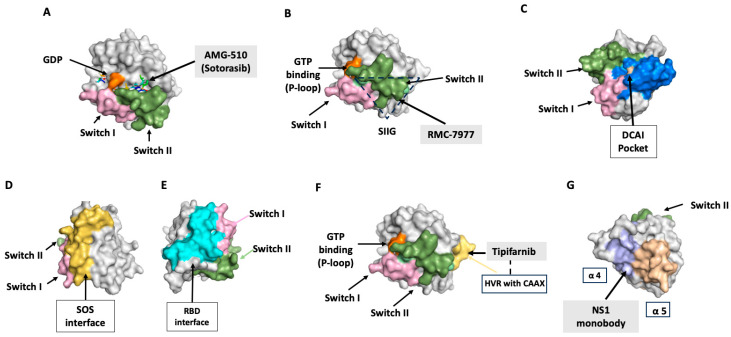
Strategies for targeting WT and mutant RAS. All structures were visualized using PyMOL, with surface models highlighting the binding interfaces; adapted from [87]. (**A**) Direct targeting of GDP-bound KRAS-G12C with covalent inhibitor AMG-510 (sotorasib), binding to the switch II pocket, in orange (PDB: 6OIM). (**B**) Direct targeting of wild-type (WT) and G12C KRAS with RMC-7977 (PDB: 4OBE). RMC-7977 binds to the switch II groove (SIIG) of RAS; (**C**) GDP-bound KRAS with the SOS1-mediated nucleotide exchange inhibitor DCAI (PDB: 4DST). The surface model highlights the DCAI pocket in yellow. (**D**,**E**). Indirect targeting of RAS: surfaces targeted by inhibitors of SOS (a RAS guanine nucleotide exchange factor, GEF, (**D**)) or effector protein binding (RAS-binding domain, RBD, (**E**)) (PDB: 6GJ8). (**F**) Targeting post-translational modification of RAS with tipifarnib (PDB: 4JV6). This structure shows KRAS in complex with farnesyltransferase and the inhibitor tipifarnib, preventing farnesylation of the HVR within CAAX motif. (**G**) Allosteric inhibition of Ras by the NS1 monobody (PDB: 5E95). The NS1 binding site is highlighted in purple.

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
