# Peer review of "Towards Targeting Endothelial Rap1B to Overcome Vascular Immunosuppression in Cancer"

_ijms, 2024, doi:10.3390/ijms25189853_

Round 1

Reviewer 1 Report

Comments and Suggestions for Authors

The authors reviewed endothelial anergy and vascular immunosuppression. However, it looks more like a hypothesis than a review.

1. The role of endothelial cells in resistance to anti-angiogenesis therapy has been studied recently. The reviewer should include the recent studies.

2. Immune cell infiltration is common in cancerous tissue. It is difficult to believe that endothelial cells form a barrier to immune cell infiltration in the tumour vasculature.

3. The review should be focused on EC anergy and vascular immunosuppression. However, the evidence was almost indirect.

Author Response

  1. The role of endothelial cells in resistance to anti-angiogenesis therapy has been studied recently. The reviewer should include the recent studies.

Response: We have included additional recent studies in our manuscript to further elaborate on the role of endothelial anergy in resistance to anti-angiogenic therapy.

  1. Immune cell infiltration is common in cancerous tissue. It is difficult to believe that endothelial cells form a barrier to immune cell infiltration in the tumour vasculature.

Response: Thank you for your comment. We understand that immune cell infiltration is common in cancerous tissues. However, numerous studies, including those we had cited (references 11-16 and 21-25), have demonstrated that endothelial cells can indeed play a significant role in regulating immune cell infiltration within the tumor vasculature. This process, referred to as vascular immunosuppression, involves endothelial cells forming a barrier that limits immune cell entry into the tumor microenvironment. To further support this, we have added additional references (17-20) from recent studies that highlight this phenomenon.

  1. The review should be focused on EC anergy and vascular immunosuppression. However, the evidence was almost indirect.

Response: Thank you for your feedback. We understand that the focus should be on EC anergy and vascular immunosuppression. While the review highlights pathways through which Rap1B is implicated in these processes, we acknowledge that the evidence presented may seem more indirect. We have discussed relevant pathways, such as TNF-α signaling and VEGF-mediated immunosuppression, which suggest Rap1B’s role in modulating EC responses. However, we recognize that further direct evidence linking Rap1B specifically to EC anergy is needed to fully support these conclusions.

Reviewer 2 Report

Comments and Suggestions for Authors

ijms-3161577

Type of manuscript: Review

Title: Towards targeting endothelial Rap1B to overcome vascular immunosuppression in cancer

Authors: Behshid Ghadrdoost Nakhchi, Ramoji Kosuru, Magdalena Chrzanowska *

This paper is a review article on targeting endothelial Rap1B to overcome vascular immunosuppression in cancer. This review paper is particularly timely, given that there are not many review articles specifically focused on the role of Rap1B in endothelial cells. Additionally, there is a need for research on mechanisms to overcome vascular immunosuppression. Therefore, I believe the value of this review article is recognized, especially since it was written in a field where not much research has been conducted. While the content of the review seems appropriate, a few additional revisions are needed in the review's writing.

[Major concerns]

1.   One of the most important aspects of writing a review article is the accurate use of terminology. In this review, the protein Ras or RAS, which is one of the key proteins, is inconsistently labeled. This inconsistency likely stems from the cited references. To avoid confusing the readers, ensure that you use only one consistent notation, either Ras or RAS, throughout the text and in the figures.

2.   One of the key points to keep in mind when writing a review paper is to avoid directly presenting the authors' research findings. If you wish to refer to their results, describe them from a neutral standpoint, as other researchers would, and provide proper citations. Directly stating the findings in the text, as done here, is inappropriate and should be corrected. Examples: Lines 18 and 327, etc.

3.   Line 23: The content in Line 23 is a sentence typically found in research articles. Please revise it to fit the format of a review paper.

4.   Abbreviations: The use of abbreviations when writing a review paper has many advantages besides simplicity of expression. To use an abbreviation, first write the abbreviation in parentheses after the full name, and then use the abbreviation from Introduction to the final Conclusion. Only in Abstract and Figure legend do it separately. If an abbreviation is not used more than twice, there is no need to define it, so please delete it.

5.   In cases where abbreviations are used within figures or tables, please list these abbreviations along with their corresponding full names in the figure legends or at the bottom of corresponding tables. If there are two or more abbreviations, arrange them in alphabetical order. When listing the full names of abbreviations, do not capitalize the first letter of each word unless they are proper nouns.

6.   Italicize the names of all human genes. Identify the genes used in the text and ensure they are accurately italicized. Ras vs. RAS.

7.   Reference section: Author should consult and peruse carefully recent issues of the journal, International Journal of Molecular Sciences (IJMS), for format and style. Also double-check the abbreviations of journal names. Examples: 1, 19, 21, 75, 107, 121, 141, 142, 163, 166, etc.

[Minor concerns]

1.   Lines 12 and 39: When writing the names of compounds within a sentence, do not use capital letters unnecessarily. Examples: ‘Vascular Endothelial Growth Factor’ should be written as ‘vascular endothelial growth factor’; Line 91: Apatinib, Axitinib, Sorafenib, Sunitinib.

2.   Line 62: Define PD-L1.

3.   Line 68: Define TNF.

4.   Line 69: Define NF-κB.

5.   Lines 83 and 87: Do not write ‘is’ in bold.

6.   Line 108: Define TILs.

7.   Line 132: Define ICIs.

8.   Line 135: PD1 and PDL1 should be written as PD-1 and PD-L1. Always use consistent terminology throughout the text.

9.   Line 240: Define HVR.

10.    Line 259: Define ECM.

11.    Line 368: Re-write ‘nucleotide- binding’.

12.    Line 540: ‘guanine nucleotide exchange factors’ had already been abbreviated as GEFs at Line 210. Therefore, just use GEFs here.

Overall, the manuscript can be considered to publication after major revision as indicated above.

Comments on the Quality of English Language

ijms-3161577

Type of manuscript: Review

Title: Towards targeting endothelial Rap1B to overcome vascular immunosuppression in cancer

Authors: Behshid Ghadrdoost Nakhchi, Ramoji Kosuru, Magdalena Chrzanowska *

This paper is a review article on targeting endothelial Rap1B to overcome vascular immunosuppression in cancer. This review paper is particularly timely, given that there are not many review articles specifically focused on the role of Rap1B in endothelial cells. Additionally, there is a need for research on mechanisms to overcome vascular immunosuppression. Therefore, I believe the value of this review article is recognized, especially since it was written in a field where not much research has been conducted. While the content of the review seems appropriate, a few additional revisions are needed in the review's writing.

[Major concerns]

1.   One of the most important aspects of writing a review article is the accurate use of terminology. In this review, the protein Ras or RAS, which is one of the key proteins, is inconsistently labeled. This inconsistency likely stems from the cited references. To avoid confusing the readers, ensure that you use only one consistent notation, either Ras or RAS, throughout the text and in the figures.

2.   One of the key points to keep in mind when writing a review paper is to avoid directly presenting the authors' research findings. If you wish to refer to their results, describe them from a neutral standpoint, as other researchers would, and provide proper citations. Directly stating the findings in the text, as done here, is inappropriate and should be corrected. Examples: Lines 18 and 327, etc.

3.   Line 23: The content in Line 23 is a sentence typically found in research articles. Please revise it to fit the format of a review paper.

4.   Abbreviations: The use of abbreviations when writing a review paper has many advantages besides simplicity of expression. To use an abbreviation, first write the abbreviation in parentheses after the full name, and then use the abbreviation from Introduction to the final Conclusion. Only in Abstract and Figure legend do it separately. If an abbreviation is not used more than twice, there is no need to define it, so please delete it.

5.   In cases where abbreviations are used within figures or tables, please list these abbreviations along with their corresponding full names in the figure legends or at the bottom of corresponding tables. If there are two or more abbreviations, arrange them in alphabetical order. When listing the full names of abbreviations, do not capitalize the first letter of each word unless they are proper nouns.

6.   Italicize the names of all human genes. Identify the genes used in the text and ensure they are accurately italicized. Ras vs. RAS.

7.   Reference section: Author should consult and peruse carefully recent issues of the journal, International Journal of Molecular Sciences (IJMS), for format and style. Also double-check the abbreviations of journal names. Examples: 1, 19, 21, 75, 107, 121, 141, 142, 163, 166, etc.

[Minor concerns]

1.   Lines 12 and 39: When writing the names of compounds within a sentence, do not use capital letters unnecessarily. Examples: ‘Vascular Endothelial Growth Factor’ should be written as ‘vascular endothelial growth factor’; Line 91: Apatinib, Axitinib, Sorafenib, Sunitinib.

2.   Line 62: Define PD-L1.

3.   Line 68: Define TNF.

4.   Line 69: Define NF-κB.

5.   Lines 83 and 87: Do not write ‘is’ in bold.

6.   Line 108: Define TILs.

7.   Line 132: Define ICIs.

8.   Line 135: PD1 and PDL1 should be written as PD-1 and PD-L1. Always use consistent terminology throughout the text.

9.   Line 240: Define HVR.

10.    Line 259: Define ECM.

11.    Line 368: Re-write ‘nucleotide- binding’.

12.    Line 540: ‘guanine nucleotide exchange factors’ had already been abbreviated as GEFs at Line 210. Therefore, just use GEFs here.

Overall, the manuscript can be considered to publication after major revision as indicated above.

Author Response

  1. One of the most important aspects of writing a review article is the accurate use of terminology. In this review, the protein Ras or RAS, which is one of the key proteins, is inconsistently labeled. This inconsistency likely stems from the cited references. To avoid confusing the readers, ensure that you use only one consistent notation, either Ras or RAS, throughout the text and in the figures.

Response: we corrected this, as requested.

  1. One of the key points to keep in mind when writing a review paper is to avoid directly presenting the authors' research findings. If you wish to refer to their results, describe them from a neutral standpoint, as other researchers would, and provide proper citations. Directly stating the findings in the text, as done here, is inappropriate and should be corrected. Examples: Lines 18 and 327, etc.

Response: It is appropriate to present one’s own findings in a review, provided they are balanced with other reports, presented transparently, and supported by citations. We have thoroughly reviewed our manuscript to ensure these conditions are met and have revised certain sections accordingly to align with these criteria.

  1. Line 23: The content in Line 23 is a sentence typically found in research articles. Please revise it to fit the format of a review paper.

Response: We revised the abstract to fit the format of a review paper.

  1. Abbreviations: The use of abbreviations when writing a review paper has many advantages besides simplicity of expression. To use an abbreviation, first write the abbreviation in parentheses after the full name, and then use the abbreviation from Introduction to the final Conclusion. Only in Abstract and Figure legend do it separately. If an abbreviation is not used more than twice, there is no need to define it, so please delete it.

Response: we made these changes, as requested.

  1. In cases where abbreviations are used within figures or tables, please list these abbreviations along with their corresponding full names in the figure legends or at the bottom of corresponding tables. If there are two or more abbreviations, arrange them in alphabetical order. When listing the full names of abbreviations, do not capitalize the first letter of each word unless they are proper nouns.

Response: we made these changes, as requested. It is customary to capitalize the names of growth factors, such as VEGFR2, and we left them unchanged.

  1. Italicize the names of all human genes. Identify the genes used in the text and ensure they are accurately italicized. Ras vs. RAS.

Response: this change was made as requested.

  1. Reference section: Author should consult and peruse carefully recent issues of the journal, International Journal of Molecular Sciences (IJMS), for format and style. Also double-check the abbreviations of journal names. Examples: 1, 19, 21, 75, 107, 121, 141, 142, 163, 166, etc.

Response: we used EndNote with IJMS style to format the references.

[Minor concerns]

  1. Lines 12 and 39: When writing the names of compounds within a sentence, do not use capital letters unnecessarily. Examples: ‘Vascular Endothelial Growth Factor’ should be written as ‘vascular endothelial growth factor’; Line 91: Apatinib, Axitinib, Sorafenib, Sunitinib.

Response: Throughout the manuscript, we corrected compound names to start will lowercase letters. However, acronyms such as Vascular Endothelial Growth Factor typically capitalize the full names of the words they represent; therefore we retained the capital letters in those instances.

  1. Line 62: Define PD-L1.

Response: this has been done.

  1. Line 68: Define TNF.

Response: this has been done.

  1. Line 69: Define NF-κB.

Response: this has been done.

  1. Lines 83 and 87: Do not write ‘is’ in bold.

Response: the change was made as requested.

  1. Line 108: Define TILs.

Response: this has been done.

  1. Line 132: Define ICIs.

Response: this has been done.

  1. Line 135: PD1 and PDL1 should be written as PD-1 and PD-L1. Always use consistent terminology throughout the text.

Response: this has been done.

  1. Line 240: Define HVR.

Response: this has been done

  1. Line 259: Define ECM.

Response: this has been done.

  1. Line 368: Re-write ‘nucleotide- binding’.

Response: this has been done.

  1. Line 540: ‘guanine nucleotide exchange factors’ had already been abbreviated as GEFs at Line 210. Therefore, just use GEFs here.

Response: this has been done.

Round 2

Reviewer 1 Report

Comments and Suggestions for Authors

How the vascular niche is altered to promote the T cell infiltration and activation, and the published molecular mechanisms, are not well reviewed in this review. How do the leaky tumor vessels limit the T cell infiltration.
Although ECs-T cell crosstalk has been reported, the exact information is needed to avoid confusion.

Reviewer 2 Report

Comments and Suggestions for Authors

I recommend the acceptance of the paper as the issues previously pointed out have been appropriately corrected